# REAL-TIME RATE CONTROL FOR TASK-AWARE VIDEO COMPRESSION USING REINFORCEMENT LEARNING

## ABSTRACT

Video encoders optimize compression for human perception by minimizing reconstruction error under bit-rate constraints. In many modern applications such as autonomous driving, an overwhelming majority of videos serve as input for AI systems performing tasks like object recognition or segmentation, rather than being watched by humans. It is therefore useful to optimize the encoder for a downstream task instead of for perceptual image quality. However, a major challenge is how to combine such downstream optimization with existing standard video encoders, which are highly efficient and popular. Here, we address this challenge by controlling the Quantization Parameters (QPs) at the macro-block level to optimize the downstream task. This granular control allows us to prioritize encoding for task-relevant regions within each frame. We formulate this optimization problem as a Reinforcement Learning (RL) task, where the agent learns to balance long-term implications of choosing QPs on both task performance and bit-rate constraints. Notably, our policy does not require the downstream task as an input during inference, making it suitable for streaming applications and edge devices such as vehicles. We demonstrate significant improvements in two tasks, car detection, and ROI (saliency) encoding. Our approach improves task performance for a given bit rate compared to traditional task agnostic encoding methods, paving the way for more efficient task-aware video compression.

## 1 INTRODUCTION

Video compression is an essential and widely studied problem (Bhaskaran & Konstantinides, 1997; Wenger, 2003; Sullivan et al., 2012; Bross et al., 2021; Kufa & Kratochvil, 2017). Most video compression algorithms are designed for preserving how a video is perceived by people. With the success of computer vision applications, many videos are used in automated systems, from autonomous drones and cars, to security cameras, and in downstream tasks, like object detection or recognition. In these cases, compression must prioritize regions relevant to the task at hand (e.g., allocating more bits to objects than to the background). We illustrate the need for a downstream task aware compression scheme in Figure 1. Basically, raw data is too expensive and existing encoders are geared to optimize video PSNR, which may "waste" bits on task-irrelevant parts.

Real-world deployment of compression systems complicates matters further. Video data must be collected in real time from devices, using low computational resources, and be usable for training various models across multiple tasks, not just for immediate inference. Furthermore, due to computational and hardware constraints, compression must be done without access to the ground truth for the downstream tasks during the encoding process. Our goal is to tackle these challenges by providing a general video compression method that can be adapted to any task, operates in real-time, imposes low computational demands on the encoding side, and requires no ground-truth labels.

Many existing approaches for task-aware compression rely on deep encoding (Lu et al., 2019). This makes them computationally expensive and unsuitable for real-time applications or resource-constrained environments. In contrast, standardized video encoders such as x264 (Merritt & Vanam, 2006), are highly efficient but are not designed for adapting compression to specific tasks in real-time. Some previous research proposed to use standardised video encoders for downstream tasks, but

usually for a specific task, and commonly employ big models before encodingShi & Chen (2020). As one example, Xie et al. (2022) perform semantic compression by applying a heavy feature extractor before encoding using a ground-truth segmentation maps. While performing well at this setup, their method requires large computation resources before encoding, can not be used for various tasks, and is not suitable for data collection.

In this paper, we propose RL-RC-DoT, a novel solution to the problem of tuning an efficient real-time video compression system to a downstream task without access to its ground truth labels during inference. Our approach integrates a lightweight network on the video encoder side of the a x264 encoder, trained to control the encoding process such that the decoded output is ideal for the task at hand. By leveraging standardized codecs, we ensure that our method is both computationally efficient and easily deployable across a range of devices. The solution allows for real-time video compression without requiring ground truth for downstream tasks.

Coping with these challenges is hard. Standardized encoders are not differentiable, making it difficult to optimize bit allocation for specific tasks. To overcome this, we introduce a reinforcement learning (RL) mechanism that controls the Quantization Parameter (QP) at the macro-block level, adjusting the bit allocation for each block of the frame dynamically. This allows us to efficiently manage the bit-rate budget while optimizing task performance over an entire sequence of video frames. Our experiments demonstrate that this approach yields significant improvements in rate-distortion trade-offs, not just for the task the encoder was trained on, but also for other related tasks, showcasing the robustness of our method. Furthermore, we demonstrate its generalizability by showing how an encoder trained on one model can improve performance for other models without additional tuning.

In summary, this paper makes the following contributions. (1) We design the first task-aware video compression method that builds on top of existing encoders and does not require solving the task during inference. (2) We show how to optimize the rate parameter of every macro-block in the frame while optimizing the performance of a downstream task on the reconstructed video under bit-rate constraints. (3) We design an architecture that outputs multiple actions, a tailored reward for this problem, and a task-prediction loss term. (4) We show improved rate-distortion trade-off for our agent on two tasks, car detection and ROI encoding with only small interference to image quality, and further show robustness to task shift, when tested on a related-but-different task than used for training.

## 1.1 Related Works

**Task-aware video compression with unrestricted compute.** Several previous studies proposed video compression methods that are aware of a downstream task. Zhang et al. (2024) explored content-specific filters to improve post-processing in video codecs, optimizing them for machine vision tasks like object detection and segmentation. Ge et al. (2024) introduced an encoder control for deep video compression that adapts to multiple tasks using a single pre-trained decoder, showing significant bit-rate improvement for object detection and tracking. Shor & Johnston (2022) highlighted the limitations of classical codecs in medical videos, proposing learned compression models to allocate more bits to medically relevant regions. Elgamal et al. (2020) presented a semantic video encoding system that enhances object detection by selectively decompressing frames in surveillance streams. Li et al. (2024) developed a distributed compression framework that adjusts to varying bandwidth in multi-sensor networks to optimize task performance. Windsheimer et al. (2024) introduced an annotation-free optimization strategy that aligns video coding with machine tasks, improving rate savings without relying on ground truth data. Additionally, While Wu et al. (2024) focused on real-time, quality-scalable video decoding, it also evaluated the codec performance on machine-based tasks.

All these approaches share a common limitation, they do not use the existing highly optimized and widely prevalent existing video compression ecosystem like the open-source x264 (Merritt & Vanam, 2006). The challenge therefore remains to design video compression systems that build on top of existing technology, but can be tuned in a-content adaptive way to a set of downstream tasks.

**Task-aware video compression with standard encoders** Another body of works does employ standardized encoders, but does not consider the inter-frame dependencies. Singh et al. (2022) and Fischer et al. (2020) optimize the CTU partitioning to improve the compression for a downstream

task. Galteri et al. (2018) uses a threshold on the saliency map to allocate more bits to important regions, while Cai et al. (2021) optimizes over the modelled relation between each block parameter and the task performance. Li et al. (2021) uses RL for optimizing macro-block QPs, but does so in each frame separately, where the sequence is defined over the sequence of macro-blocks in the same frame. In our work we output all macro-block QPs with one policy and the sequence is defined over consecutive encoded frames in the video. The work most related to ours is Xie et al. (2022), where they propose to use RL on both the QPs and macro-block QPs in a hierarchical manner. However, they limit their optimization to only two frames in every GOP, and only two values of macro-block QPs are chosen per block according to a given segmentation map. In our work we optimize over all frames and macro-block QPs, and we do not use any additional information like saliency, segmentation or downstream task during inference.

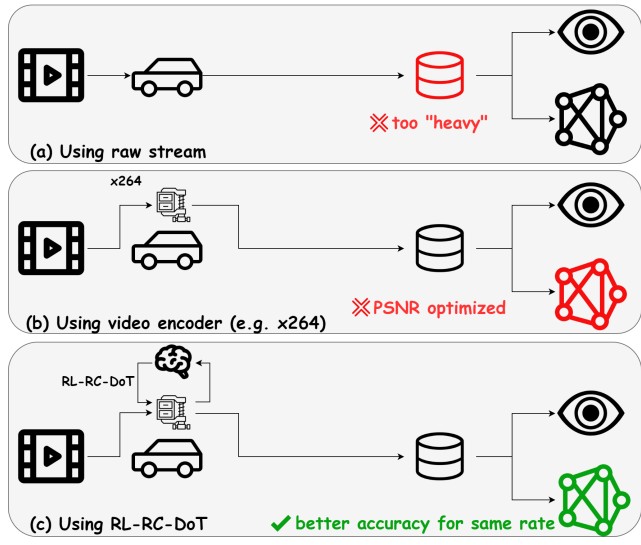

Figure 1: Approaches for video compression on a device. Videos are collected on a device (left), transmitted to a server (right) and processed. (a) Raw data transmission preserves all information but requires excessive bandwidth and storage. (b) Traditional encoding using x264 is optimized for PSNR. It reduces data size but does not prioritize task-relevant information. (c) Our RL-RC-DoT, balances task performance and bit-rate constraints.

## 2 PRELIMINARIES

### 2.1 VIDEO COMPRESSION

Video compression is a process of reducing the size of digital video files while maintaining acceptable visual quality. It is a crucial technology in the modern digital age, enabling efficient storage, transmission, and streaming of video content across various platforms and devices. The primary goal of video compression is to eliminate redundant and less perceptible information from the video data according to constraints such as bit-rate of the target video, while maintaining good visual quality.

One key aspect of video compression is the use of Quantization Parameters (QP). QP values control the level of compression applied to the video data, with higher values resulting in more compression but lower quality, and lower values preserving more detail but producing larger file sizes. In video encoding, QP can be applied at different levels of granularity. Frame QP refers to setting a single QP value for an entire frame, which is useful for maintaining consistent quality across the frame but may not be optimal for all areas. Per-macro-block (MB) QP, on the other hand, allows for finer control by assigning different QP values to individual MBs within a frame, usually in small perturbations from a pre-assigned frame QP. This approach enables the encoder to apply more compression to less important or visually complex areas while preserving quality in critical regions. Per-MB QP can lead to more efficient compression and better overall visual quality, as it adapts to the local characteristics of the video content. It is especially suitable for task-aware optimization since most tasks target specific areas in the picture (for instance object detection and segmentation).

The effectiveness of video compression is typically measured by comparing the compressed video's file size and visual quality to the original. Metrics like Peak Signal-to-Noise Ratio (PSNR) and Structural Similarity Index (SSIM; Wang et al. (2004)) are often used to objectively assess quality. When comparing two encoders the compression efficiency is usually considered. To do so, a video

is encoded in several desired bit-rates with each encoder to form a rate-distortion (RD) curve, where the $y$ axis is the quality measure, usually PSNR. If one encoder's curve is higher than the other, it means it suffers less distortion for the same bit-rate rendering it more efficient. If we integrate over the entire curve, and average the result over multiple videos, we obtain a quantity specifying how much more efficient one encoder than the other, a quantity referred to as Bjontegaard delta rate (BD-rate) (Wiegand et al., 2003).

With the increasing usages of videos for machine vision, many researchers have recognized the need for task-aware compression and proposed a suitable evaluation metric (Kong et al., 2016; Shi & Chen, 2020). The most straightforward metric which we also use in this paper is obtained by replacing the PSNR in the RD-curve (the $y$-axis) with a task-specific loss measure such as mIOU or detection precision and calculating the BD-rate with respect to the adjusted curves.

One may wonder, if a downstream task is given, why is video compression needed at all? For instance, in the autonomous vehicle example, if a car detector is available, why not run that detector on the vehicle, and save only its decision instead of the compressed video. There are several strong reasons not to take this approach: (1) Many downstream tasks require resource-heavy networks that cannot run efficiently on-device, making it impractical to process the data locally. (2) Sending only task-specific features limits human interpretability, as there would be no watchable video for explainability. (3) This also confines the data to a single task, preventing its reuse for other applications or analyses. (4) Large-scale data collection, such as in autonomous driving, depends on compressed video storage; using features alone would limit future training and fine-tuning opportunities. (5) Task-specific features are often tied to a particular model, making them incompatible with new models, while compressed video remains adaptable across different systems. We show that our method allows different models to achieve high performance using the same compressed data. This is also the reason why we aim to develop a method that still preserves a video that would be meaningful to a person.

## 2.2 REINFORCEMENT LEARNING

Reinforcement Learning (RL; Sutton & Barto (1998)) is a field dealing with sequential decision making in unknown environments. To formulate a problem using RL, we first need to define its underlying Markov Decision Process (MDP). An MDP is defined by a tuple $(\mathcal{S}, \mathcal{A}, P, R, \gamma)$, where $\mathcal{S}$ is a finite set of states, $\mathcal{A}$ is a finite set of actions, $P$ is a state transition probability function, $P(s'|s, a)$, $R$ is a reward function, $R(s, a)$ and $\gamma \in [0, 1]$ is a discount factor.
At each time step $t$, the agent observes the current state $s_t \in \mathcal{S}$ and chooses an action $a_t \in \mathcal{A}$. The environment then transitions to a new state $s_{t+1}$ with probability $P(s_{t+1}|s_t, a_t)$ and the agent receives a reward $r_t = R(s_t, a_t)$. The goal of the agent is to find a policy $\pi : \mathcal{S} \to \mathcal{A}$ that maximizes the expected cumulative discounted reward:

$$\max_{\pi} J^{\pi} = \mathbb{E}_{\pi, s_0 \sim \mu, s_{t+1} \sim P}\left[\sum_{t=0}^{\infty} \gamma^t R(s_t, \pi(s_t))\right]$$

To do so, many algorithms were proposed in the literature varying in their assumptions on the problem, computational complexity and data requirements. Perhaps the most widely used algorithm today is PPO (Schulman et al., 2017) which directly optimizes the policy using full trajectories while constraining it from diverging.

## 3 METHOD

We present RL-RC-DoT, an **RL**-based **R**ate **C**ontroller for **Do**wnstream **T**ask, that dynamically optimizes macro-block QP deltas during video encoding. To formalize the training framework, we cast the video compression problem with respect to a downstream task as an MDP. We define the state of the environment to be block-wise statistics extracted from x264 MB-tree mode (Garrett-Glaser, 2009) (block energy cost, inverse quantization scaling factor, etc.) and global statistics (bit-stream size, percentages of P blocks, etc.). For a detailed list of the statistics used as state, see Appendix A.1.

The action space is defined as the choice of all macro-block QP deltas within a frame. For instance, given a frame resolution of 480×320 pixels partitioned into 16×16 pixel macro-blocks, the resulting

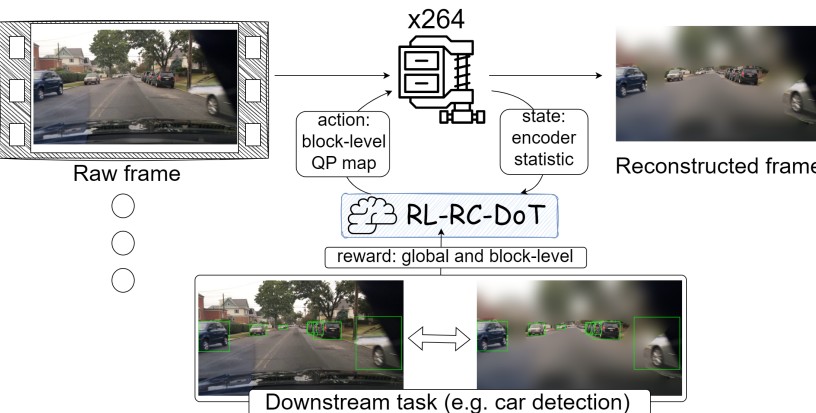

Figure 2: **RL-RC-DoT workflow.** Our proposed solution to the block-level control for a downstream task. RL-RC-DoT takes encoder statistics as input and outputs a block-level delta QP map. We then evaluate the difference in downstream task performance between the reconstructed frame and the raw frame. The reward contains both global score as reward and block-level score.

action space constitutes a 30×20 dimensional matrix, where each value represent the delta between the frame level QP and the block specific QP. However, the convergence of reinforcement learning algorithms on high-dimensional action spaces presents significant computational challenges. To address this limitation, we implement a hierarchical approach: during the learning phase, we operate on a lower-resolution action space, which is subsequently upsampled to the original dimensions through interpolation. This dimensionality reduction technique facilitates more efficient training while maintaining the ability to generate fine-grained QP assignments. We analyze the impact of action space resolution on model performance in Appendix A.5. A diagram of the full system is given in Figure 2.

We define the reward as a combination of two rewards with different purposes. First, we want to ensure compliance with the encoder's bit-rate constraint. This is particularly crucial in streaming applications, where exceeding the allocated bandwidth can result in frame dropping and consequently deteriorate the viewer experience. The reward component $r_{\text{bit-rate}}$ for this objective is defined as:

$$r_{\text{bit-rate}} = -\left|\log\left(\frac{\text{current average bit-rate}}{\text{target bit-rate}}\right)\right|.$$

The second objective is to maximize the performance of the downstream task on the decoded frames. Since ground-truth data is unavailable, we introduce a novel self-supervised approach. This method treats the downstream task's output on the original uncompressed frame as a pseudo-ground-truth, against which we evaluate the task performance on the reconstructed frame:

$$r_{\text{DT}} = D(f_{\text{DT}}(\text{frame}_{\text{raw}}), f_{\text{DT}}(\text{frame}_{\text{rec}})),$$

where $f_{\text{DT}}$ is a pre-trained model for the downstream-task and $D$ is a task-specific loss function. For example, in the context of vehicle detection, $f$ is a pre-trained car detection model (e.g. YOLO-v5 nano (Jocher, 2020)), and $D$ is the precision between $f(\text{frame}_{\text{rec}})$ with respect to $f(\text{frame}_{\text{raw}})$. Finally we use the weighted reward $r = r_{\text{bit-rate}} + \lambda r_{\text{DT}}$ for some hyper-parameter $\lambda$, in order to optimize the rate-performance trade-off.

### 3.1 MACRO-BLOCK REWARD INFORMATION

In most RL problems, the reward is a black-box directly mapping the state to a continuous score. Recent literature (Ye et al., 2021) has demonstrated that predictive modeling of rewards – implemented as auxiliary heads alongside policy or value networks can significantly enhance agent performance. In our setup, the reward presents a unique characteristic: the reward signals for various downstream tasks are often compositional, derived from aggregating scores across granular components of the input frame. For example, in the case of saliency-weighted PSNR, the reward is computed by aggregating per-pixel reconstruction errors. Leveraging this decomposable nature of rewards, we

propose augmenting the learning process with an auxiliary prediction loss for these sub-scores during backpropagation. Specifically, we introduce a block-wise prediction loss that aims to predict the individual block reward information that contribute to the overall task score. This approach of incorporating auxiliary prediction loss for macro-block level reward information is expected to enhance the agent's performance. Firstly, it provides a more granular learning signal, allowing the agent to understand the impact of its actions on individual components of the reward. Secondly, by learning to predict these sub-scores, the agent develops a richer internal representation of the task structure. Lastly, this method aligns the agent's learning more closely with the actual composition of the reward, potentially leading to faster convergence and more stable learning. We show the effect of this improvement in Section 5.4.

## 4 EXPERIMENTS

We evaluate our approach with two downstream tasks: car detection and region of interest encoding (Liu et al., 2008). We further study the robustness of the method, when a trained compression policy is tested with a different car detector, or even in a segmentation task instead of detection. Finally, we report performance of ablation experiments.

### 4.1 DATASET

We trained and evaluated RL-RC-DoT using a subset of video streams from the BDD100K dataset (Yu et al., 2020), a large-scale driving video dataset, with multi-task annotations. We reconstructed the raw data from the videos and to allow faster training time, we resized them to a smaller resolution of 480x320 pixels. We then filtered out streams that exhibited trivial rate-task performance (RD) curves with respect to the downstream tasks of car detection precision, when encoded with the standard x264 codec (Wiegand et al., 2003). We specifically excluded streams that showed zero precision across most target bit-rates. This approach ensured that our dataset presented meaningful challenges for compression optimization. Our final dataset comprised of 172 streams in total, with 65 streams used for training our agent, 7 streams used for evaluation on different hyper-parameters and 100 streams reserved for testing. For reproducibility, we provided a detailed list of the specific stream used in our experiments in appendix A.2 of this paper.

### 4.2 EVALUATION METRICS: RD-CURVE AND BD-RATE

Since compression is a constraint optimization problem, it is standard to depict results using a Rate-Distortion (RD) curve. An RD-curve illustrates the trade-off between bit-rate constraint and quality in video compression (see examples in Figure 3). RD-curves are traditionally used with PSNR, but are equally applicable to task-specific metrics like precision/recall for a detection task or saliency-weighted PSNR for ROI-based encoding. These RD-curves allow us to evaluate compression efficiency for any downstream tasks on reconstructed videos.

BD-rate (Bjøntegaard Delta rate; (Bjøntegaard, 2001)) is a widely used metric in video compression to compare the efficiency of different encoding methods. This method calculates the average difference in bit-rate between two rate-distortion (RD) curves at the same quality level. The BD-rate represents the percentage of bit-rate savings that one encoding method achieves over another while maintaining equivalent video quality performance. Therefore, a negative BD-rate indicates that the test method requires less bits than the reference method to achieve the same quality / task performance.

### 4.3 EXPERIMENTAL DETAILS

All our experiments use the x264 open source encoder software (Merritt & Vanam, 2006). we used *medium* preset and target bit-rates $50 - 200$ kbps. To extract the MB-tree statistics we allow x264 to use look-ahead for 10 frames. For car detection, we employ YOLOv5-nano (Jocher, 2020). ROI encoding is evaluated using saliency maps generated by TranSalNet (Lou et al., 2022). Our agent is trained agent using PPO implemented in Stable-Baselines3 (Raffin et al., 2021) with corresponding reward function as described in Section 3. We augment the standard PPO algorithm with a reward per block prediction network, as described in Section 3.1. To facilitate efficient training, we utilize 8

parallel environments running on an Intel(R) Xeon(R) CPU E5-2698 v4 @ 2.20GHz, complemented by an NVIDIA Tesla V100 32GB GPU. Each agent undergoes training on 20 million frames, a process that spans approximately 4 days. Our training achieves a frame rate of roughly 50 FPS, while evaluation in a single environment maintains around 30 FPS, demonstrating the feasibility of real-time streaming applications.

**Compared methods:** To conduct a fair and meaningful comparison against existing baselines, baseline should be solving the same task, and particularly have access to the same information. Several previous studies developed methods for task-aware encoding, but their setup is fundamentally different. For instance, Xie et al. (2022) assume that compression has access to the output of a segmentation module for each frame during inference. Li et al. (2021) and Fischer et al. (2020) focus on single-frame (image) compression, without considering the overall video budget constraints. Finally, most methods did not release code Shi & Chen (2020); Fischer et al. (2020). These differences in approach and constraints would make direct comparisons potentially misleading.

## 5 RESULTS

### 5.1 CAR DETECTION

We first assess the performance of RL-RC-DoT in the context of video compression optimized for car detection. The reward function for training our RL agent is based on the precision score of YOLOv5-nano (Jocher, 2020). For our additional auxiliary loss described in Section 3.1, we compute the precision score for each individual block separately to generate block-specific reward information. After training, we evaluate the policy on 100 test videos from the BDD100K dataset.

Table 1 compares RL-RC-DoT with the standard x264 encoder, focusing on the detection performance of the YOLOv5-nano detector on compressed videos. The evaluation is conducted across multiple compression rates, with results averaged over all frames in the test dataset for each target bit-rate. We also applied the same comparative approach to assess the PSNR of the reconstructed streams. The results demonstrating that RL-RC-DoT improves car detection precision and recall significantly, with minimal impact on the PSNR of the compressed videos.

Figure 3 illustrates the superiority of RL-RC-DoT over the standard x264 encoder through RD curves for three representative video streams. Figure 4, shows a qualitative example of the performance gain. We compare the images in both types of rate-control, and the output of the downstream task. We can see the details corresponding to the downstream task are better reconstructed yielding a more relevant image.

To quantify the performance difference between methods, we compute the BD-rate (see 4.2), a standard metric in the field. Our approach shows significant improvements in detection performance, with BD-rate reductions of $24.7\%(\pm1.38\%)$. These gains come at a minimal cost to overall video quality, yielding a slight increase (deterioration) in PSNR BD-rate of $1.19\%(0.46\%)$ (3). This means that videos compressed using RL-RC-DoT remains understandable to human viewers. This aspect is crucial for validation and debug purposes. It also provides robustness to changes of task models, a point we elaborate on in section 5.3. The relatively small bit-rate error indicates that the encoder maintains its ability to adhere to the requested bit-rate for the entire video.

### 5.2 ROI ENCODING

| | Precision | | Recall | | PSNR | |
|---|---|---|---|---|---|---|
| | low-rate | high-rate | low-rate | high-rate | low-rate | high-rate |
| x264 | $.22 \pm .0013$ | $.66 \pm .0015$ | $.45 \pm .002$ | $.81 \pm .001$ | $28.98 \pm .03$ | $34.55 \pm .03$ |
| RL-RC-DoT(ours) | $\mathbf{.36\pm .0015}$ | $\mathbf{.71 \pm .0014}$ | $\mathbf{.63 \pm .002}$ | $\mathbf{.83\pm .001}$ | $29.03 \pm .03$ | $34.55 \pm .03$ |

Table 1: Car detection precision and recall of YOLO5, and PSNR. Value are mean and s.e.m. calculated across all frames from a test set of 100 videos from BDD100K.

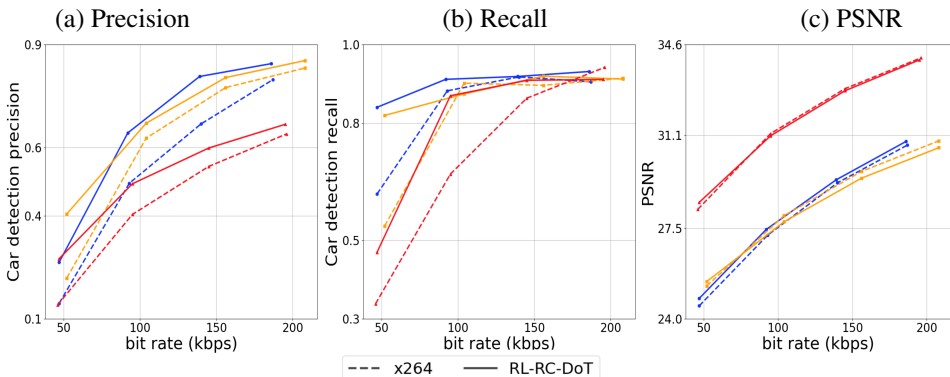

Figure 3: Rate-Quality curves for Car detection task. Comparing standard x264 (dashed lines) with RL-RC-DoT (solid lines). Curves show three example streams, demonstrating how RL-RC-DoT improves quality across the range of bit-rate values. **(a)** Car detection precision **(b)** recall **(c)** PSNR.

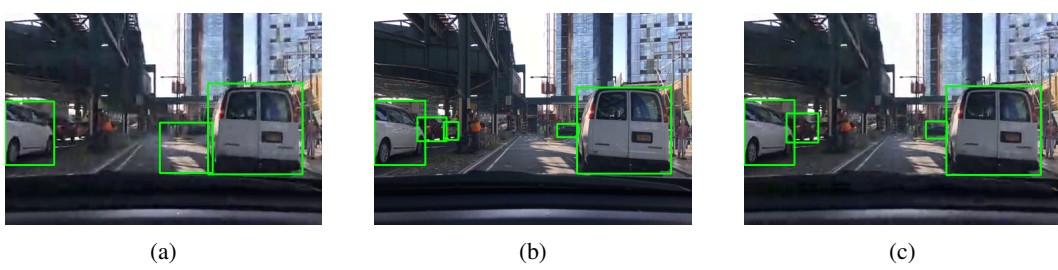

Figure 4: Car detection example result. (a) detection output on x264 reconstructed frame, (b) output on raw frame and (c) output on RL-RC-DoT reconstructed frame

We conducted similar experiments for ROI-encoding task by promoting saliency weighted PSNR as the task score. RL-RC-DoT demonstrates significant improvements in saliency-weighted PSNR encoding efficiency on the test-set, as shown in Table 2. RL-RC-DoT exhibits a BD-rate value of $-25.64\%(\pm 0.99\%)$, indicating that our method achieves better quality in salient regions at lower bit-rates compared to x264. Interestingly, the PSNR BD-rate is slightly better than the vanilla rate-control. This may be due to the proximity between the two tasks. This also shows the sub-optimality of the vanilla rate-control when considering specific content. Finally, RL-RC-DoT achieves similar average bitrate error as x264.

Figure 5 illustrates the Rate-Distortion (RD) curves for three representative video streams. These curves demonstrate that in most cases, RL-RC-DoT achieves a more favorable RD trade-off for ROI encoding task compared to x264. Figure 6 provides qualitative examples of our method's performance, visually illustrating the enhanced quality in salient regions compared to the baseline encoding.

## 5.3 TASK ROBUSTNESS

An important concern is that RL-RC-DoT might overfit for the training task. That would mean that changing the model, may harshly hurt performance. We set to evaluate robustness to such changes in RL-RC-DoT by training the policy with one downstream task ,and testing it with another.

| ROI encoding experiment | Saliency-weighted PSNR BD-rate | PSNR BD-rate | Bit-rate error $[1e-3]$ |
|---|---|---|---|
| RL-RC-DoT | $-25.64 \pm 0.99$ | $-5.26 \pm 0.36$ | $-1.0 \pm 0.43$ |

Table 2: Results on RL-RC-DoT applied on the test-set for the saliency-weighted PSNR task.

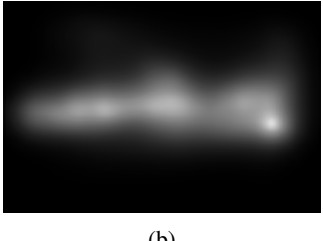
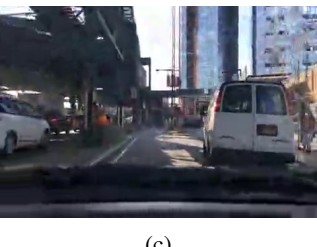

(a)                              (b)                              (c)

Figure 6: Saliency weighted PSNR results. (a) x264 reconstructed frame, (b) Saliency map of raw frame, extracted with  (Lou et al., 2022) (c) RL-RC-DoT reconstructed frame.

|  | Precision (YOLO) | Recall (YOLO) | PSNR | Precision (SSD) |
|---|---|---|---|---|
| RL-RC-DoT | $-24.7 \pm 1.57$ | $-19.75 \pm 2.97$ | $1.19 \pm 0.46$ | $-26.2 \pm 1.48$ |
|  | **Recall (SSD)** | **Segmentation IOU** | **Bit-rate error** $[1e-3]$ | |
| RL-RC-DoT | $-25.81 \pm 2.03$ | $-14.6 \pm 1.81$ | $0.13 \pm 0.44$ | |

Table 3: BD-rate Results on RL-RC-DoT applied on test set for the car detection task for various settings. Negative values mean that RL-RC-DoT improves over baseline.

More specifically, we optimized the policy for car detection using the YOLOv5-nano model, as described in section 5.1. Then, we measured the detection performance of another model, SSD (Liu et al., 2016). We also measure the performance on the related but distinct task of car segmentation (DeepLab; (Chen et al., 2017)). The results are also listed in Table 3.

This approach allows us to examine whether our method truly captures fundamental aspects of visual information relevant to automotive perception tasks, rather than overfitting to a specific model or narrow task definition. By demonstrating performance improvements across different models and related tasks, we aim to show that our compression method preserves task-relevant information in a more general sense, potentially allowing for model updates or task modifications without the need to retrain the compression policy. This robustness is crucial for real-world applications where deployed systems may need to adapt to new models or slightly different tasks over time.

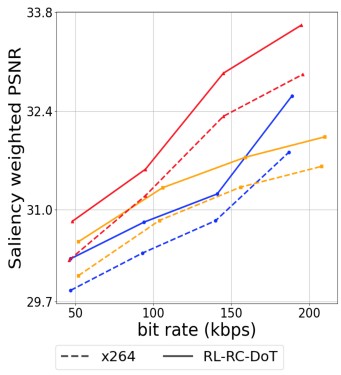

Figure 5: RD-curves for 3 videos for ROI-encoding.

For car detection evaluated with SSD, the precision BD-rate is very similar to precision with YOLOv5-nano, which was used for training. For car segmentation, although tested with a different task, we still observe an improved but weaker BD-rate than the detection task. This improvement can be attributed to the close relation between the tasks, so meaningful macro-blocks for car detection, are also useful for the segmentation task. In summary, the BD-rate obtained on the PSNR and the various tasks show the robustness of our method to new tasks and new models that solve the task.

Figure 7, shows a qualitative example of task robustness. We compare the images in both types of rate-control, and the output of the downstream task. We can see the details corresponding to the downstream task are better reconstructed yielding a more relevant image.

## 5.4 ABLATION EXPERIMENTS

To quantify the relative contribution of various components of our method, we perform ablation studies, for both car detection and ROI encoding, and provide the results in Table 4. For both tasks,

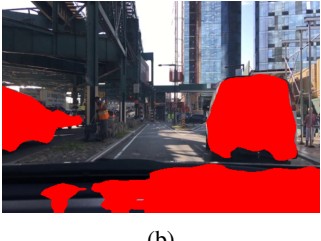 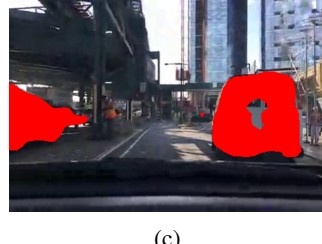

|  (a)  |  (b)  |  (c)  |

Figure 7: Car segmentation result comparison. (a) segmentation output on x264 reconstructed frame, (b) output on raw frame and (c) output on RL-RC-DoT reconstructed frame

| Car detection | Precision BD-rate | Recall BD-rate | PSNR BD-rate | Bit-rate error $[1e-3]$ |
|---|---|---|---|---|
| RL-RC-DoT | $-24.7 \pm 1.57$ | $-19.75 \pm 2.97$ | $1.19 \pm 0.46$ | $0.13 \pm 0.44$ |
| RL-RC-DoT w/o RI | $-19.4 \pm 1.38$ | $-11.94 \pm 1.7$ | $1.92 \pm 0.41$ | $0.4 \pm 0.47$ |
| RL-RC-DoT $\gamma = 0$ | $-9.78 \pm 1.29$ | $-10.28 \pm 2.14$ | $5.44 \pm 0.6$ | $2.4 \pm 0.44$ |

| ROI encoding | Saliency-weighted PSNR BD-rate | PSNR BD-rate | Bit-rate error $[1e-3]$ |
|---|---|---|---|
| RL-RC-DoT | $-25.64 \pm 0.99$ | $-5.26 \pm 0.36$ | $-1.0 \pm 0.43$ |
| RL-RC-DoT w/o RI | $-23.46 \pm 0.97$ | $-4.54 \pm 0.42$ | $5.3 \pm 0.48$ |
| RL-RC-DoT $\gamma = 0$ | $-16.01 \pm 0.77$ | $2.11 \pm 0.31$ | $6.9 \pm 0.43$ |

Table 4: Ablation study. (1) Full RL-RC-DoT (2) Omitting reward information (RI) from the training process and (3) Ignoring long term effects by using a myopic policy.

we first ablated the macro-block reward information as described in Subsection 3.1. Then, ran an experiment for $\gamma = 0$ which shows what happens when optimizing for a myopic policy.

The results show that reward info improved the learning process and reduces the BD-rate even further for both tasks. This demonstrates the benefit of exploiting additional information in the video compression domain that is generally not available. For $\gamma = 0$, the BD-rate is significantly worse for both tasks. As expected, ignoring the future implications of the bit-allocation can cause sub-optimal decisions for the entire video. This also emphasizes the limitation of rate-control methods optimizing for every frame separately; a common practice by previous works. Finally, the small PSNR BD-rate shows that the encoder does not drastically reduce the picture quality, and the requested bit-rate is preserved as evident by the low bit-rate error (native encoder bit-rate error is $-2.9 \cdot 10^{-3}$).

## 6 CONCLUSIONS AND LIMITATIONS

Machine learning for videos understanding became prevalent in numerous applications, but impose high costs of storing, making fast encoding and low bit-rate critical. Task-aware compression has huge potential, but existing methods have critical limitations, like heavy compute or dependency on ground truth task data for compression. We develop an efficient RL solution which encodes every frame in real time while optimizing the future bit-rate and task performance on the reconstructed video. Our learned policy is robust against changes in the downstream models for the same task and to closely related tasks, showing large important potential for data collection for autonomous vehicle, patient monitoring and robotics.

**Limitations:** Training our models involves encoding and performing the downstream task per frame, and this may slow down converge depending on the complexity of the downstream task. Also, generalizing across video resolutions may be hard because it affects the size of action space and the complexity of the learning problem.

## 7 REPRODUCIBILITY STATEMENT

**Encoder environment:** To apply rate-control on the environment we changed the code of the open source x264 (Merritt & Vanam, 2006) encoder so that in each frame it can obtain delta-QP values externally and provide relevant statistics as described in Appendix A.1.

**RL Agent:** We provide a description of the policy's architecture in Appendix A.3. The agent was trained using PPO implementation from stable-baselines3 (Raffin et al., 2021) with default parameters, where we just added an MSE prediction loss (with weight 0.1) for reward info. We used $\lambda = 20$ to average between the bit-rate and downstream task rewards.

**Experiments:** In our experiments we used the publicly available BDD100K dataset (4.1) which was resized using the open source package ffmpeg. We provide the named list of streams we used in Appendix A.2. In the experimental details subsection 4.3 we provide additional information on the hardware we used and the downstream task models we used for our experiments.

## 8 ETHICS STATEMENT

Our method's approach to selectively altering the quality of different regions within a frame raises important considerations regarding the perceptual integrity of the reconstructed video. By optimizing compression for specific downstream tasks, there is a potential risk of introducing unintended perceptual biases or distortions that may not be immediately apparent to human viewers.

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

# A    APPENDIX

## A.1    ENVIRONMENT DETAILS

| Global encoder statistics used as state information |
|:---:|
| Next frame x264 selected QP value |
| Next frame number |
| Current bitstream size |
| Current frame x264 selected QP value |
| Average QP |
| Percentages of I type Macro Blocks |
| Percentages of P type Macro Blocks |
| Percentages of skip-type Macro Blocks |
| x264 calculated PSNR |
| x264 calculated SSIM |
| Percentages of bits used for Motion Vectors |
| Percentages of bits used for DCT coefficient |
| Progress of encoding |
| Next frame type |
| Next frame complexity |

Table 5: Detailed components of global encoder statistic used in state information

| Local (per-MB) encoder statstic used as state information |
|:---:|
| x264 energy values per Macro Block |
| x264 intra encoding cost per Macro Block |
| x264 propagating encoding cost per Macro Block |
| x264 inverse quantization scale factor per Macro Block |

Table 6: Detailed components of per-MB encoder statistic used in state information

## A.2    BDD100K STREAMS

Here we elaborate on the streams we used from bdd100k dataset (Yu et al., 2020):

| Train streams | |
|---|---|
| 0000f77c-6257be58 | 000e0252-8523a4a9 |
| 000f157f-dab3a407 | 000f8d37-d4c09a0f |
| 00a04f65-af2ab984 | 00a0f008-3c67908e |
| 00a0f008-a315437f | 00a1176f-0652080e |
| 00a1176f-5121b501 | 00a2e3ca-5c856cde |
| 00a2e3ca-62992459 | 00a2f5b6-d4217a96 |
| 00a395fe-d60c0b47 | 00a9cd6b-b39be004 |
| 00abd8a7-ecd6fc56 | 00abf44e-04004ca0 |
| 00adbb3f-7757d4ea | 00afa5b2-c14a542f |
| 00afa6b9-4efe0141 | 00b04b30-501822fa |
| 00b1dfed-a89dbe2b | 00be7020-457a6db4 |
| 00beeb02-ba0790aa | 00c12bd0-bb46e479 |
| 00c29c52-f9524f1e | 00c41a61-4ba25ad4 |
| 00c497ae-595d361b | 00c87627-b7f6f46c |
| 00ca8821-db8033d5 | 00cb28b9-08a22af7 |
| 00ccf2e8-59a6bfc9 | 00ccf2e8-ac055be6 |
| 00ccf2e8-f8c69860 | 00ce6f6d-50bbee62 |
| 00ce8219-12c6d905 | 00ce8219-d0b5582e |
| 00cef86b-204ea619 | 00cef86b-d8d105b9 |
| 00cf8e3d-3d27efb0 | 00cf8e3d-4683d983 |
| 00cf8e3d-773de15e | 00cf8e3d-a7b4978c |
| 00d0f034-6d666f7b | 00d18b13-52d3e4c4 |
| 00d4b6b7-7d0a60bf | 00d4b6b7-a0b1a3e0 |
| 00d7268f-fd4487be | 00d79c0a-23bea078 |
| 00d79c0a-a2b85ca4 | 00d84b1d-21e6fe01 |
| 00d8944b-e157478b | 00d8d95a-74aa476a |
| 00d9e313-7d75bb18 | 00d9e313-926b6698 |
| 00dc5030-237e7f71 | 00de601c-858a8a8d |
| 00de601c-cfa2404b | 00e49ed1-9d41220c |
| 00e4cae5-c0582574 | 00e5e793-f94de032 |
| 00e81dcc-b1dd9e7b | 00e8c106-e197c4b1 |
| 00c50078-6298b9c1 | 00b93c6e-6298aa25 |
| 0000f77c-cb820c98 | |

Table 7: List of streams used in training

| Validation streams | |
|---|---|
| 00d8d95a-47d98291 | 00e02d60-54df99d1 |
| 00a820ef-d655700e | 00ce95b0-84be34a3 |
| 00d15d58-9197cde54 | 00b04b12-a7d7eb85 |
| 00c17a92-d4803287 | |

Table 8: List of streams used in validation

| Test streams | | | |
|---|---|---|---|
| cd35ea13-f49ee278 | cd389564-8be2128e | cdc05b0a-3bb83a9c | cd389564-9053f5fc |
| cd3b1173-63cb9e2e | cd3dab20-1b3e564e | cd3dab20-4ea3d971 | cd3df92f-d04e142c |
| cd40cb21-18170d03 | cd4ac25c-61a9eb11 | cd4bf816-2abb75c9 | cd4bf816-c2f9bf78 |
| cd4ce4e5-6994fd2d | cd4ce4e5-d0968ec0 | cd4da443-da4fe8c7 | cd4deee2-0703d1c7 |
| cd4deee2-1d9539bd | cd4deee2-37c8b95c | cd4deee2-3feadd6e | cd4deee2-60291439 |
| cd4deee2-688c8bba | cd4deee2-8e12e5b5 | cd4deee2-9c9f6da1 | cd4deee2-adc7e92a |
| cd4deee2-ce4f69f5 | cd4deee2-d078d54a | cd547736-3b63cb96 | cd583365-462cca17 |
| cd5a94cf-345f214a | cd5a9e1b-86faac85 | cd5b2540-465c9328 | cd5b2540-913cb8f7 |
| cd5bee17-bef4f177 | cd5db4e0-1189ff83 | cd6af452-e54a1e36 | cd6c087e-03ca2127 |
| cd6fdd33-ac9cb2db | cd704168-1231930e | cd7c12c7-7029da5d | cd7c12c7-9b46c2a8 |
| cd7c92a7-3b20257f | cd7c92a7-89b23268 | cd7c92a7-9222ee19 | cd7c92a7-ed0d3926 |
| cd7ee0b1-dd286a1b | cd7fb8f1-3d347a66 | cd828461-db8b4612 | cd839842-cd859db0 |
| cd8b00aa-4aac0701 | cd8b00aa-5c017145 | cd8b00aa-f00ad3b9 | cd8b30b0-51369077 |
| cd8b30b0-e8d12cc4 | cd8d2fde-2d2a3211 | cd9b6b86-9f62a970 | cd9b6b86-be582832 |
| cd9cd3dd-d67bf5b6 | cd9d84d4-f59d3feb | cd9dff27-94731aba | cd9e7e2b-4b274850 |
| cda33556-28510da1 | cda33556-8dc294b4 | cda33556-c6b3dd45 | cda55704-362ddfea |
| cda55704-754aac99 | cda63e8d-0afbf52b | cda63e8d-76b2fa43 | cda9acc1-1a92349d |
| cda9acc1-4469e473 | cda9acc1-9d1ef61a | cdac4037-afed765d | cdac7315-fe37a1d9 |
| cdae6e60-0fb06a75 | cdae6e60-334ffc87 | cdae6e60-b729f2e6 | cdaee377-1eccb13a |
| cdaee377-2263611a | cdaee377-2b38ae2c | cdb06fa9-cfb70e11 | cdb06fa9-eba5643a |
| cdb3b01b-673f85b7 | cdb616df-393f382c | cdb688d4-33f24ca3 | cdb6b049-c96359c8 |
| cdb815da-d03b9395 | cdb992be-f0f1613c | cdbb20a9-bdab1f4e | cdbbac37-49c0a335 |
| cdbc7842-b72c4915 | cdbd1882-bdd416ea | cdbeedfd-4ab64af8 | cdbf4bd1-0c65ed7a |
| cdc05b0a-3bb83a9c | cdc05b0a-c53c36a6 | cdc05b0a-c6e8b6ec | cdc05b0a-ce908cf7 |
| cdc05b0a-d4ff800b | cd3dab20-1b3e564e | cdc05b0a-efb78be5 | cdc05b0a-f2a67b44 |

Table 9: List of streams used in test

### A.3 AGENT ARCHITECTURE

To train the policy, we use the PPO algorithm (Schulman et al., 2017), where the architecture of the policy is as follows: The per-block statistics are processed through a compact convolutional neural network (CNN) comprising three convolutional layers. These layers employ kernel sizes of $3x3$ or $4x4$ with a stride of 1. The resulting features are subsequently flattened and concatenated with the global statistics. A fully connected layer then derives a latent representation of dimension $64$. This latent representation serves as input to three distinct fully connected networks: the value network (critic), the policy network (actor), and the reward prediction network described in the following subsection. A diagram of the full system is given in Figure 2.

### A.4 TASK ACCURACY TO DISTORTION TRADE-OFF

As previously discussed, RL-RC-DoT gains BD-rate reductions of $24.7\%(\pm1.38\%)$ with respect to car detection precision task, while paying a minimal cost to overall video quality, as evidenced by a slight increase in PSNR BD-rate of $1.19\%(0.46\%)$. This is important since we want video to still be watchable by human eyes, for validation purposes and robustness to changing task models.

To further illustrate this point, in Figure 8 we show the PSNR and task performance BD-rate obtained by RL-RC-DoT for each stream in the test set. In the plots we see the PSNR varies around $0$ while the tasks performance is well below.

### A.5 ACTION SPACE RESOLUTION

Since we show our results on a videos of size 480x320 with macro-blocks of size 16x16, the action space is of size 30x20. The size of the action space drastically affects the performance of the agent and the convergence rate of the training process. Thus, we propose to set a lower resolution action space and upsample to the original action space by interpolation. The trade-off here is clear – if

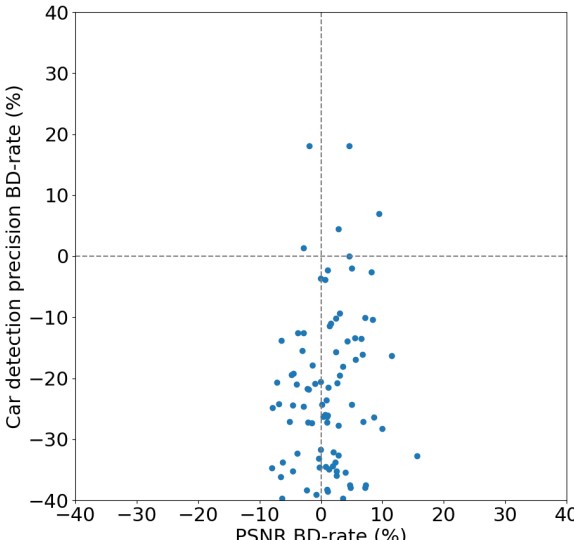

Figure 8: PSNR BD-rate to detection precision BD-rate, where each point represent a single stream in the test set

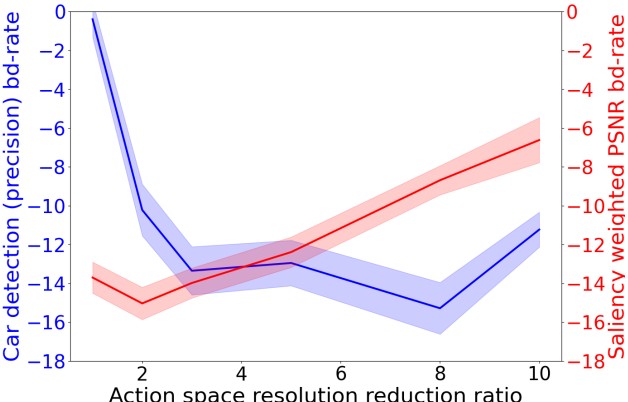

Figure 9: The effect of action space resolution on the BD-rate for both tasks

we make decisions in high resolution, the agent can take a long time to converge, whereas a low resolution decision will not provide the finer control required for accurate bit allocation for the downstream-task resulting in a sub-optimal performance. We illustrate this notion in Figure 9. We plot the task BD-rate for multiple choices of resolution reduction ratios for each of the tasks. The plot indeed shows the trade-off between the two, where each task has a different optimal choice for action space resolution. We note that these results may depend on the number of frames allotted for training, where we expect longer training to benefit lower resolutions.

