# OpenReview forum: "Real Time Macro-Block Rate Control for Task-Aware Video Compression Using Reinforcement Learning"
_ICLR.cc/2025/Conference — ICLR 2025 Conference Withdrawn Submission_

### Official Review · Reviewer_a2Py · 2024-11-04

**Soundness:** 1
**Presentation:** 1
**Contribution:** 1
**Rating:** 3
**Confidence:** 5

**Summary:**

This paper introduces a reinforcement learning-based approach to task-aware video compression. The proposed method learns an RL agent to decide a QP map for each video frame coded by x264 in order to achieve task-aware video compression. Notably, the state signals are the MB-tree statistics with 10-frame look-ahead.

**Strengths:**

(1) The idea of using RL to perform task-aware video compression is interesting.

(2) The proposed method is tested on two tasks: object detection and ROI-based coding.

**Weaknesses:**

(1) The paper is lengthy and verbose. Lots of space is used to introduce some very basic concepts of video coding and quality metrics.

(2) It appears that collecting the state signals is very costly. But, this part is not addressed at all.

(3) The idea of introducing an RL agent for task-aware video compression is NOT new.

(4) There are a few missing references.

(5) There are many technical details that need further clarification.

**Questions:**

(1) In Section 3, it is unclear what it means by “D is the precision between f(frame_rec) and f(frame_raw)”. Do you measure the distortion between frame_rec and frame_raw in the latent domain of the recognition network? Or this refers to the precision difference resulting from the replacement of the original image with the compressed image. If the task is ROI-based coding, how is D evaluated?

(2) About the macroblock-level reward, it is unclear how the r_bit_rate is evaluated for one specific macroblock.

(3) The action space is all the MB QP deltas within a frame, and is reduced by downsampling. I wonder if the agent can generalize to processing videos of different resolutions.

(4) By looking at Tables 5 and 6 in the appendix, I realize that it is very costly to get the state signals since collecting some of them may require pre-encoding and pre-decoding the input video, e.g. intra encoding cost per MB, x264 calculated PSNR and SSIM, percentages of bits used for MV, etc. It is also mentioned that the MB-tree statistics were collected by allowing x264 to use look-ahead for 10 frames. In this case, how come the evaluation could achieve 30FPS?

(5) The training and test split is kind of weird: 65 streams for training and 100 for testing.

(6) In terms of PSNR, it is unclear why the rate-distortion performance of RL-RC-DoT is comparable to that of x264. From your training objective, there is no guarantee that this would be true.

(7) Likewise, there appears to be no guarantee that the proposed agent would produce generalizable decoded videos.

(8) It is unclear whether separate agents need to be trained in order to produce multiple rate-distortion points.

(9) For the ROI task, it appears that the agent must be very capable in order to predict the weighted sum of r_bit_rate and r_DT and to identify the ROI regions, or the state signal must be very informative. Is the same network architecture used across different tasks? Fig. 6 alone is not conclusive.

(10)	There are missing references.
Ho et al., "Neural Frank-Wolfe Policy Optimization for Region-of-Interest Intra-Frame Coding with HEVC/H.265,"  VCIP’22.
Ho et al., "A Dual-Critic Reinforcement Learning Framework for Frame-level Bit Allocation in HEVC/H.265 ," DCC'21.

---

### Official Review · Reviewer_Vd76 · 2024-11-04

**Soundness:** 3
**Presentation:** 3
**Contribution:** 2
**Rating:** 5
**Confidence:** 4

**Summary:**

This paper introduces a method that automatically adjusts compression ratio of a video in a task aware fashion, specifically for object detection and semantic segmentation tasks. The proposed method formulates a region-specific compression ratio control as a Markov decision process (MDP), then uses reinforcement to train an agent that chooses the most sensible Quantization Parameters (QP) at the block level in standard H.264 encoding. Evaluation on BDD-100 videos demonstrate stronger rate-distortion tradeoff when measured in the target tasks, while having roughly the same visual quality compared to standard H.264 encoding.

**Strengths:**

## Presentation
I find the paper well-written with sufficient illustrations of all the technical concepts. In particular, it has provided a thorough literature survey on task-aware video compression methods, from which a strong motivation of the design choices made to build the presented methodology is presented.

## Novelty
This approach is novel in the following sense: (1) It is a method that directly controls parameters in the H.264 encoding. Unlike many prior works in task-aware compressions that uses deep encoding, this designs makes the proposed a real-time method. (2) Compared to prior works based on H.264 encoding, this work can model long-term dependency by formulating the problem as Markov decision process. (3) The method preserves human-viewable videos after compression, unlike some prior works, making it easier to use the compressed data for purposes beyond the task it was trained on, e.g. for further analysis, training or transfer to related tasks that the agent is not trained on.

I think the method is particularly interesting in its exploration to use H.264 encoder information as states without resorting to any deep features. It is a pretty interesting choice, in the following sense. (1) By using the H.264 encoder states it does greatly simplify the MDP making it easier to learn an effective video compression policy. (2) It is a pretty smart engineering choice as H.264 encoder itself is content aware but its processing is highly optimized. The proposed method essentially hit a sweet spot that makes full use of the feature extraction capability of H.264 encoder, instead of re-creating the wheels as is in many deep encoding based video compression schemes.

## Empirical results
The results demonstrates the following convincing evidence that this approach is indeed promising.
1. Section 5.1 and Section 5.2 demonstrates that the proposed method can reach higher precision/recall of both the car detection task and ROI encoding task compared to vanilla H.264 counterparts under the same bit constraints. Better still, it does so without sacrificing the PSNR metrics, meaning the compressed videos are roughly as watchable as vanilla H.264.

2. Section 5.3 shows that the proposed method can generalize to related tasks that are not seen during training, and still yielding much stronger results compared to vanilla H.264 encoding in terms of task accuracy.

3. In the ablation of Section 5.4 the proposed MDP formulation is shown to improve rate-distortion tradeoff compared to a myopic policy (a.k.a. a one-step decision model), which shows the method is indeed learning to make better compression ratio decision based on long-term information.

**Weaknesses:**

The main limitation of this work is in the baselines it is compared with.

- The main baseline used in this work is the vanilla H.264 encoder, without the control on macro-block QP parameter. This is a natural choice in the sense that the proposed method is built on top of a standard H.264 encoder, with the only significant change being the way the macro-block QP parameters are selected. However, given the many prior works discussed in the related work section, it seems necessary to make some comparisons with prior works, especially the most relevant "task-aware video compression with standard encoders" discussed in Line 106 - Line 118.

- While the work itself has not imposed limitations on the type of tasks it can handle, it was only tested on car detection, ROI encoding (saliency) and semantic segmentation. While these are important practical tasks, the decision making process of these tasks appear to be fairly localized - one can tell whether a car is a car from a local patch. Given that, one may wonder whether this approach can be applicable to vision tasks where the semantics are more complex, i.e. tasks that require encoding of global context. This may well expose weaknesses in this work, especially when compared with methods based on deep encoding, as it will become harder to solely rely on built-in mechanisms in H.264 encoders to determine what regions to keep.

- The object detection method tested in this work is YOLO. One of the main motivations of this work is to avoid running the detectors on-device due to resource constraints. However, YOLO can achieve good performance on many embedding devices, which seems to defeat the stated purpose of the work. In a sense, this work would be most useful in cases where an expensive detector (or counterparts for other tasks) is needed, but such a case is not really tested. Note that it can make a difference. YOLO as a fast detector may mis-detect or create false positives in certain regions with motion blurs, complex semantics etc.., which may not be the case for a better but slower detector. If the proposed methods are only good at finding those cases, it will be less performant with other expensive detectors.

**Questions:**

It would be useful to address the issues discussed in the weaknesses section. Briefly, I think it can be summarized as follows.

- The comparison with any existing task aware compression schemes, not just the H.264 baseline.
- Performance on tasks that requires some semantic reasoning in a global level.
- Results on modern, high performance detectors, such as DINO.

---

### Official Review · Reviewer_s3J3 · 2024-11-04

**Soundness:** 3
**Presentation:** 3
**Contribution:** 3
**Rating:** 5
**Confidence:** 3

**Summary:**

This paper tackles task-aware video compression. Instead of emphasizing the PSNR, the proposed aims at compressing the video while making sure that the compressed video can derive good downstream task performance. The proposed proceeds by combining downstream optimization with existing standard video encoders, which are widely used and efficient. Specifically, they address this challenge by training an RL that controls the Quantization Parameters at the macro-block level, corresponding to local patches, e.g., which patch should be compressed more than the others. By doing so, the RL can learn the optimal policy to predict the QPs to be applied to each patch. Also, to improve the efficiency, the authors proposed a multi-level strategy. The method is tested on two tasks, object detection and segmentation, and the authors show solid improvements.

**Strengths:**

Video compression is important for application domains, where processing resources and storage are limited. The proposed leverages RL to learn effective policies that control the compression quality based on the task needs. The proposed framework is easy to understand and the usage of existing encoders and off-the-shelf algorithms ensures reproducibility. The performance shows reasonable tradeoffs between the image quality and the task performance.

**Weaknesses:**

Besides the conceptual simplicity and the effectiveness of object detection and segmentation, the proposal has several weaknesses.
1) The proposed is only evaluated on two tasks, which emphasize more on the good locality of the compression. In other words, the task is not high-level, or leveraging global statics, which in turn means that if the policy understands where the objects are and can compress more on the non-object regions, then the performance should be good. With that being said, the task is reduced to lightweight object detection. The authors should justify that the derived method does not just perform simple tasks like this. Also, the authors should test on more high-level tasks as this will help us understand the applicability of the proposed.
2) The macro-block framework is also a limiting factor, e.g., a too-coarse decomposition of the image may result in suboptimal solutions, while a fine-grained one can result in better solutions, but result in difficulties in training and inference. A multi-level scheme has been proposed, but it is not clear how this multi-level optimization can be generalized to other tasks.

**Questions:**

The major question about the proposed method is how the task information is incorporated during the test. I am actually by the description in line 025, where you said the proposed does not need task information in inference. But if you only test on object detection, and the policy is trained for objection, then there is no need for task information anymore during training, as it is known by default. However, in the introduction, it seems that the authors emphasize the importance of the awareness of the task information during compression. Can you elaborate more on the underlying logic?

---

### Official Review · Reviewer_ayMQ · 2024-11-04

**Soundness:** 3
**Presentation:** 1
**Contribution:** 1
**Rating:** 3
**Confidence:** 4

**Summary:**

This article proposes a rate control algorithm for video compression for machine vision tasks formulated as a reinforcement learning problem. The agent receives block-level vectors with encoding statistics and predicts the optimal quantization parameter (QP). The reward combines minimizing rate and maximizing a task-specific score (in this case supervised for detection by the boxes predicted by a detector on the original uncompressed image). The experiments show improved performance in the evaluated tasks compared with the baseline at the same rate.

**Strengths:**

- The experiments demonstrate the utility of the approach in several tasks compared to the (task-agnostic) baseline.
- The encoding process doesn't require a network to extract auxiliary semantic information such as bounding boxes or semantic maps.

**Weaknesses:**

* **Novelty.** My main concern is the limited novelty compared to previous works such as (Li et al 2021) and its conference version (Shi and Chen 2020), which also propose block-level reinforcement learning algorithms for task-aware video compression based on QP control. The main difference I perceive is not conditioning the agent on extracted semantic information, which is a minor modification, in my opinion. Other minor modifications are the use of an object detection instead of segmentation as task, and perhaps different block statistics as input vector. The description in the paper doesn't provide enough details to assess possible differences in formulation and implementation with respect to those works. But overall, I think the novelty is very limited.
* **Presentation.**
  * Figures. Figures 1 and 2 are too abstract and uninformative. Figure 2 in particular should include more details about module architectures and representations (e.g. see Figures 1 and 2 of (Li et al 2021) for reference).
  * Lack of details in the description of the method. The authors should include a more clear and precise description of the implementation of their method. In particular, the mathematical formulation should be more precise and detailed up to the level of blocks, which are the main units of processing by the agent.
* **Experiments.** There is no comparison with relevant baselines. In particular to (Li et al 2021), which has code available at https://github.com/USTC-IMCL/Task-driven-Semantic-Coding-via-RL.

**Questions:**

Please refer to the weakenesses section for suggestions and concerns, and kindly address them in the response. In particular, clarify the questions about novelty and comparison to baselines.

---

### Note · Authors · 2024-11-13

I have read and agree with the venue's withdrawal policy on behalf of myself and my co-authors.